# Purification Process and In Vitro and In Vivo Bioactivity Evaluation of Pectolinarin and Linarin from *Cirsium japonicum*

**DOI:** 10.3390/molecules27248695

**Published:** 2022-12-08

**Authors:** Yana Ye, Zhenlin Chen, Yonglin Wu, Mengmeng Gao, Anqi Zhu, Xinyuan Kuai, Duosheng Luo, Yanfen Chen, Kunping Li

**Affiliations:** 1Institute of Chinese Medicine Sciences, Guangdong Pharmaceutical University, Guangzhou 510006, China; 2Key Laboratory of Glucolipid Metabolic Disorder, Ministry of Education of China, Guangzhou 510006, China; 3Guangdong TCM Key Laboratory for Metabolic Diseases, Guangdong Pharmaceutical University, Guangzhou 510006, China; 4School of Traditional Chinese Medicine, Guangdong Pharmaceutical University, Guangzhou 510006, China

**Keywords:** *Cirsium japonicum*, pectolinarin, lipopolysaccharide, acute liver injury, metabolomics

## Abstract

Pectolinarin and linarin are two major flavone O-glycosides of *Cirsium japonicum,* which has been used for thousands of years in traditional Chinese medicine. Pharmacological research on pectolinarin and linarin is meaningful and necessary. Here, a process for the purification of pectolinarin and linarin from *C. japonicum* was established using macroporous resin enrichment followed by prep-HPLC separation. The results show the purity of pectolinarin and linarin reached 97.39% and 96.65%, respectively. The in vitro bioactivities result shows the ORAC values of pectolinarin and linarin are 4543 and 1441 µmol TE/g, respectively, meanwhile their inhibition rate of BSA-MGO-derived AGEs is 63.58% and 19.31% at 2 mg/mL, which is 56.03% and 30.73% in the BSA-fructose system, respectively. The COX-2 inhibition rate at 50 µg/mL of linarin and pectolinarin reached 55.35% and 40.40%, respectively. Furthermore, the in vivo bioassay combining of histopathologic evaluation and biochemical analysis of liver glutamic oxaloacetic transaminase, serum creatinine and TNF-α show pectolinarin can alleviate lipopolysaccharide (LPS)-induced acute liver and kidney injury in mice. Metabolomics analysis shows that pectolinarin attenuates LPS-challenged liver and kidney stress through regulating the arachidonic acid metabolism and glutathione synthesis pathways. Collectively, our work presents a solid process for pectolinarin and linarin purification and has discovered a promising natural therapeutic agent—pectolinarin.

## 1. Introduction

*Cirsium japonicum* Fisch. ex DC is a kind of medicinal plant listed in the Chinese Pharmacopoeias, which has been used in Traditional Chinese medicine for thousands of years [1]. Some recent studies have shown that the clinical effects of *C. japonicum* are related to their high contents of flavonoids, including two main flavone O-glycosides, pectolinarin and linarin (Figure 1) [2,3]. Consequentially, pharmacological research on pectolinarin and linarin is meaningful and necessary [4]. In fact, pectolinarin and linarin have been reported to have antihemorrhagic, antidiabetic, hepatoprotective and antitumor bioactivities [5], but more solid evidence is urgently needed for their therapeutic potential evaluation. Now, adequate pectolinarin and linarin are needed for further pharmaceutical and molecular mechanism research. However, up to now, knowledge on the process for the preparation of high-purified natural pectolinarin and linarin from bench-top to higher scales is limited.

The isolation and purification of natural compounds is an arduous task, and a series of techniques and processes have been developed. In recent years, macroporous resins have been successfully applied for the separation of various types of natural products [6], such as flavonoids, alkaloids, and phenolic compounds, with many advantages, such as low cost and reusability. However, the traditional process of the adsorption/desorption of macroporous resins cannot meet the high purity standard, so it should be combined with other techniques to harvest the target compounds. Preparative HPLC (prep-HPLC) is a chromatographic technique that has been widely used for the separation of natural products due to its many advantages of online monitoring, efficient separation, and automatic control. However, it typically requires pre-treatment to remove excessive impurities, which may contaminate the chromatographic columns [7]. Considering the possible complementarity between macroporous resins and prep-HPLC, a process combining them was established for the rapid separation and purification of target compounds from *C. japonicum.*

In addition to the purity of pectolinarin and linarin, their bioactivities are also critical concerns of the target compounds. Reactive oxygen species (ROS) are products produced by all aerobic organisms in the process of conventional oxygen metabolism [8]. Excessive ROS production leads to increased oxidative stress in vivo, which can induce various chronic diseases, such as metabolic disorders, cardiovascular disease, and cancer [9,10]. Antioxidants are an essential defense against oxidative stress damage. Therefore, the search for effective antioxidants is crucial for the prevention of various chronic diseases. Similarly, inflammation involves a series of complex defense-related reactions caused by various damage factors [11]. A balance is maintained between pro-inflammatory and anti-inflammatory cytokines in normal physiological conditions [12]. However, inflammatory response dysregulation has been proven to disrupt system homeostasis and result in the development and progression of various diseases, including rheumatoid arthritis, atherosclerosis and nonalcoholic fatty liver disease [13]. Therefore, the control of inflammation is of great significance to reduce the occurrence of long-term tissue damage and diseases. In addition, advanced glycation end products (AGEs) are the products of the glycation reaction, which is a spontaneous, naturally occurring non-enzymatic reaction that occurs between the aldehyde group of reducing sugars and the amino groups of proteins, nucleotides, and lipids [14]. The long-term intake of food rich in AGEs will lead to the accumulation of AGEs in the circulation system, which will promote the development and progression of a series of chronic diseases, such as Alzheimer’s disease, diabetes, cardiovascular disease, kidney failure, retinopathy and so on [15,16]. Briefly, the progress of many chronic diseases is related to oxidative damage, AGEs and inflammation. However, many potent synthetic inhibitors for oxidative damage, AGEs and chronic inflammation have been given increasing attention due to their side effects. Therefore, the screening and developing of novel natural compounds is meaningful and necessary.

It will be potentially fruitful to screen the lead compounds from those herbal drugs with long histories of human usage [17,18]. In the present study, two main flavone O-glycosides of *C. japonicum*, namely, pectolinarin and linarin, were purified by macroporous resins enrichment followed by prep-HPLC separation, and their in vitro and in vivo activities were also evaluated. Our results present an innovative process for the effective purification and screening of natural products with three main effects, namely, antioxidant, anti-AGEs and anti-inflammation activities, which should also be useful for some other natural products’ research.

## 2. Results and Discussion

### 2.1. Macroporous Resin Enrichment Process

#### 2.1.1. Selection of Macroporous Resins

The separation characteristics of macroporous resins depend in large part on their specific area, surface polarity and pore diameters [19]. Either non-polar resins or polar macroporous resins are applicable for the adsorption of flavonoids or phenolic acids, but as far as specific target components are concerned, the adsorption/desorption capacity of resin must be taken into accounted. As shown in Table 1, 8 kinds of macroporous resins, including three polar (resins NKA-9, HPD-600 and HPD-500), two weak polar resins (AB-8 and DM 130), and three non-polar resins (D-101, HP-20 and HPD-100) were tested for their adsorption/desorption capacity for TFC. In the present experimental condition, the total flavonoid content was calculated using a standard curve (Y = 13.175X + 0.0124), and the linear correlation coefficient was *R*^2^ = 0.9992. The data show that the D-101 macroporous resin had the highest desorption capacity of TFC among the eight tested resins, while the adsorption capacity of D-101 resin for TFC was lower than those of HPD500 and AB-8 resins. This can be explained as the larger average pore size can facilitate the desorption process, thereby possibly increasing the desorption capacity. As we all know, the chemical composition of the crude aqueous extract of *C. japonicum* is complicated, and there are also some flavone aglycones except for flavone glycosides. Compared with AB-8 resin, D-101 resin has better desorption efficiency for flavone O-glycoside, including pectolinarin and linarin [20]. Therefore, D-101 resin was selected for further purification because of its large surface area and ideal average pore diameter, which correlate with the chemical features of the adsorbate molecules [21].

#### 2.1.2. Adsorption Kinetics of Resins

The kinetics of adsorption are amongst the key properties that describe the efficiency of adsorption. Figure 2A shows the adsorption kinetics of the TFC at 25 °C. The adsorption of the TFC progressed rapidly in the first 100 min and then slowed down. After 150 min, the adsorption capacity of the TFC gradually reached equilibrium. This trend reveals that the adsorption of the TFC on D-101 was a fast process. Therefore, the adsorption time must exceed 150 min to ensure complete adsorption of the TFC. To better comprehend the mechanistic aspects of the adsorption of TFC on D-101, two classic models describing pseudo-first-order and pseudo-second-order were utilized to fit the experimental data. The equations describing each of these kinetic models are given below [22].

Pseudo-first-order kinetic:$$\ln\left(Q_{e}-Q_{t}\right)=lnQ_{e}-K_{0}t$$

Pseudo-second-order kinetic:$$\frac{t}{Q_{t}}=\frac{1}{K_{1}Q_{e}{}^{2}}+\frac{t}{Q_{e}}$$
where *K*_0_ and *K*_1_ represent the rate constant for pseudo-first-order models and pseudo-second-order models. *Q_e_* and *Q_t_* are the adsorption capacities at equilibrium and at time *t*, respectively. 

The characteristic parameters of adsorption kinetics are shown in Table 2. The value of *R*^2^ for the pseudo-second-order kinetic model was higher than that obtained from fitting with the pseudo-first-order kinetic model. The value of *Q_e_* for the pseudo-second-order kinetic model was more reasonable than that for the pseudo-first-order kinetic model compared to the experimental data. Thus, the pseudo-second-order model proved to be the most suitable for the adsorption of the TFC on D-101 resin, and was applied to analyze the complete adsorption process. Further, the pseudo-second-order kinetic model operates under the assumption that the adsorption concentration of adsorption is closely related to the rate-limiting step in the adsorption process. In other words, the adsorption of the TFC to D-101 resin may be of a chemical nature. For the constant of the dynamic model of the TFC adsorption, the values of *K*_1_ were positive, which indicates that the *Q_e_* values for the TFC adsorption on D-101 resin increased with adsorption time until equilibrium was established, as validated by Figure 2A.

#### 2.1.3. Adsorption Isotherms on Resins

Adsorption isotherms are an important reference for the reasonable selection of adsorbent varieties for specific purposes. The adsorption isotherms of TFC on D-101 resin were constructed based on the data originated from the adsorption experiments at 20, 30 and 40 °C. As shown in Figure 2B, the higher the temperature was, the weaker the adsorption capacities of *C. japonicum* on D-101 resin. This result indicates that the adsorption process is an exothermic reaction [23]. Additionally, when the initial concentrations of aqueous solutions of crude *C. japonicum* extract increased, the adsorption capacities also increased. The classic Langmuir and Freundlich isotherms, used to fit the experimental data, are represented by the equations below [24].

Freundlich adsorption model:$$\ln Q_{e}=\frac{1}{n}\ln C_{e}+\ln K_{F}$$

Langmuir adsorption model:$$\frac{C_{e}}{Q_{e}}=\frac{1}{K_{L}Q_{m}}+\frac{C_{e}}{Q_{m}}$$
where *K_F_* is the Freundlich constant and 1/*n* denotes the equation exponent related to the adsorption driving force. *Q_m_* (mg·g^−1^) represents the saturated adsorption capacity, and *K_L_* (mL·mg^−1^) is the Langmuir equation constant.

The values of all parameters of the Langmuir and Freundlich models are listed in Table 3. The correlation coefficients of fitting to the Langmuir isotherm were higher than those for the Freundlich isotherm, indicating that the Langmuir isotherm is a good model for the adsorption of TFC on D-101 resin. When the adsorption temperature increased from 20 to 40 °C, the *Q_m_* (mg·g^−1^) value increased from 34.24 to 56.75 mg·g^−1^, indicating that the increase in temperature enhanced the adhesion power of D-101 resin to TFC. The value of *n*, a measure of adsorption intensity or surface heterogeneity, determines the shape of the adsorption isotherm. The value of *n* is above 1, which proves that the adsorption process is favorable to adsorption, and the value of *n* is below 1, which indicates that the adsorption process is cooperative with adsorption [25]. In this experiment, all values of *n* are above 1, indicating that the adsorption process of TFC on D-101 resin is favorable.

### 2.2. Purification and Identification of Pectolinarin and Linarin

#### 2.2.1. HPLC Analysis

The HPLC chromatograms of *C. japonicum* extracts before and after D-101 resin enrichment are shown in Figure 3A,B. It can be observed that the relative peak areas of total flavonoids increased after enrichment by the resin. In addition, many water-soluble and polar impurities that exhibit no ultraviolet absorption due to the lack of conjugated groups were removed using nonpolar macroporous resins. Hence, the chromatographic peaks of select impurities removed by the resin may not be observed in the chromatograms of Figure 3A,B.

#### 2.2.2. Purification and Confirmation of Linarin and Pectolinarin

As described in the Section 3.3, a process for the further purification of high-purity linarin and pectolinarin was proposed using prep-HPLC after enrichment by D-101 resin. Shown as Figure 3C,D, the purity of the resulting linarin (Compound **1**) and pectolinarin (Compound **2**) reached approximately 96.65% and 97.39%. The purity meets the requirements of various bioassays. The spectrum data are listed in the following (See detail in Appendix A).

Compound **1**, linarin, was isolated as a white powder. Its molecular formula was determined as C_28_H_32_O_14_ based on the HR-ESI-MS spectrum with a pseudo molecular peak discovered at *m*/*z* 591.1651. The NMR data were as follows: ^1^H-NMR (500 MHz, DMSO-d_6_) δ: 12.91 (1H, s, 5-OH), 8.05 (2H, d, 9.0 Hz, H-2′/6′), 7.15 (2H, d, 9.0 Hz, H-3′/5′), 6.94 (1H, s, H-3), 6.79 (1H, d, 2.1 Hz, H-8), 6.45 (1H, d, 2.1 Hz, H-6), 5.07 (1H, d, 7.3 Hz, H-1″), 4.55 (1H, d, 1.7 Hz, H-1‴), 3.86 (3H, s, 4′-OCH3), 1.08 (3H, d, 6.3 Hz, H-6‴); ^13^C-NMR (126 MHz, DMSO-d_6_) δ:182.0 (C-4), 164.0 (C-2), 163.0 (C-7), 161.1 (C-5),157.0 (C-8a), 105.5 (C-4a), 99.7 (C-6), 94.8 (C-8),103.8 (C-3), 162.4 (C-4′), 128.5 (C-2′,6′), 122.7 (C-1′), 114.7 (C-3′,5′), 100.5 (C-1″), 75.7 (C-5″), 76.3 (C-3″), 73.1 (C-2″), 69.6 (C-4″), 66.1 (C-6″), 100.0 (C-1‴), 72.1 (C-4‴), 70.8 (C-3‴), 70.4 (C-2‴), 68.3 (C-5‴), 17.8 (C-6‴), 55.6 (-OCH_3_). Compound **1** was then identified as linarin by comparing the above spectral data with the literature [26]. 

Compound **2**, pectolinarin, was isolated as a white powder. Its molecular formula was determined as C_29_H_34_O_15_ based on the HR-ESI-MS spectrum with a pseudo molecular peak discovered at *m*/*z* 621.1754. The NMR data were as follows: ^1^H-NMR (500 MHz, DMSO-d6) δ: 8.04 (d, J = 9.1 Hz, 2H, H-2′, H-6′), 7.17 (d, J = 9.0 Hz, 2H, H-3′, H-5′), 6.94 (s, 2H, H-3, H-8), 5.12 (d, J = 6.8 Hz, 1H, H-1”), 4.56 (s, 1H, H-1‴), 3.86 (s, 3H, OMe), 3.77 (s, 3H, OMe), 1.05 (d, 3H, H-6‴); ^13^C-NMR (126 MHz, DMSO-d_6_) δ: 182.4 (C-4), 164.1 (C-2), 162.4 (C-4′), 156.6 (C-9), 152.3 (C-7), 152.2 (C-5), 132.7 (C-6),128.5 (C-2′, C-6′), 122.7 (C-1′), 114.8 (C-3′, C-5′),105.9 (C-10), 103.4 (C-3), 100.4 (C-1‴, C-1”), 94.4 (C-8), 76.5 (C-3”), 75.6 (C-5”), 73.2 (C-2”), 72.0 (C-4‴), 70.8 (C-3‴), 70.5 (C-2‴), 69.5 (C-4”), 68.3 (C-5‴), 66.0 (C-6”), 60.1 (6-OMe), 55.6 (4′-OMe), 17.8 (C-6‴). Compound **2** was then identified as pectolinarin by comparing the above spectral data with the literature [2].

### 2.3. Bioactivity Assay In Vitro

#### 2.3.1. ORAC Assay

ORAC values are primarily used to assess antioxidant capacity via the rate of fluorescence decay. In the present study, the fluorescence decay curves of Trolox (from 12.5 μM to 200 μM) and different compounds are shown in Figure 4A–C. The final ORAC values were calculated using the regression equation for the Trolox concentration plotted against the net area under the fluorescence decay curve (AUC; area under the curve). Data are expressed as ORAC value (µmol Trolox Equivalent/g of sample; µmol TE/g). The regression equations of the Trolox standard curve are Y = 17.306x + 50.817, *R*^2^ = 0.9966. The ORAC assay revealed that the ORAC value of pectolinarin was up to 4543 µmol TE/g, which is higher than that of linarin (1441 µmol TE/g), indicating that pectolinarin had a better antioxidant activity (Figure 4D). The occurrence of the oxidation process is related to the presence of excess free radicals, and in general, the free radical-trapping antioxidant activity of flavonoids is related to the transfer of hydrogen atoms to peroxy radicals [27]. This can be used to show that pectolinarin has better free radical inhibitory activity than linarin because it can donate hydrogen atoms.

#### 2.3.2. Anti-AGEs Assay

AGEs have been proposed to be a candidate biomarker of a series of chronic diseases, such as Alzheimer’s disease, diabetes, aging, and so on [28]. New interventions are always expected for preventing or reducing the negative impact of AGEs-related damage. So, as described in the methods Section 3.4.2, the antiglycation capabilities of pectolinarin and linarin were evaluated by measuring the relative contents of fluorescent AGEs. In the BSA-MGO-derived AGEs formation model, after 7 days of incubation, the inhibition rates of pectolinarin and linarin at the concentration of 2 mg/mL were 63.58% and 19.31%, respectively (Figure 5A). Similarly, the inhibition rates of pectolinarin and linarin were 56.03% and 30.73% in the BSA-fructose-derived AGEs formation model (Figure 5B). Although both of them are not better than the positive control, aminoguanidine, as they were 91.91% and 73.24%, respectively, pectolinarin should be taken into account as a potential lead compound. In addition, under both AGEs-formation systems, the inhibition potential of pectolinarin is higher than linarin, although there is just one more methoxy group substituted on the C-5 position in the A-ring of the canonical flavone backbone. Their structure–activity relationships maybe an interesting issue that is beyond the scope of this research. Furthermore, the inhibition activity of the two compounds in the BSA-MGO model was higher than that in the BSA-fructose model, which suggests that the two compounds may be more selective in targeting MGO than other α-dicarbonyl compounds to stop the glycation progress [29].

#### 2.3.3. COX-2 Inhibition Assay

The cyclooxygenase-2 (COX-2) is a key enzyme in the synthesis of prostaglandins, and its inhibitors are effective for the treatment of inflammation. Our data show that in the analysis of inhibiting COX-2, at the same concentration of 50 µg/mL, the inhibition rate of linarin reached 55.35%, which was higher than that of pectolinarin (40.40%) (Figure 6A). However, as the present study was just a preliminary screening, the IC_50_ values of these two compounds were not measured, otherwise the inhibitory activity between them could be better compared. Overall, we confirm that both the antioxidant and anti-AGEs activities of pectolinarin were better than those of linarin, while the anti-COX-2 activity of linarin was better than pectolinarin. Molecular docking is mainly used to explore the binding mode of proteins and compounds. The binding energies of pectolinarin and linarin are 247.70 and 44.06 kcal/mol, respectively. The molecular docking results reveal that a number of conventional hydrogen bond and carbon hydrogen bond interactions were predicted to be established between pectolinarin and the side chains of MET552, TYR355, LEU384 and ALA527. The linarin was predicted to interact with the side chain of LEU384, which is also an amino acid that interacts with pectolinarin. Moreover, GLY516, TYR385 and PHE518 can form Amide-Pi stacked interactions with pectolinarin, and TRP387 and TYR385 can form Pi-Pi T-shaped interactions with linarin.

According to the aforementioned results, the adsorption kinetics curve of crude extract of *C. japonicum* on the D101 resin was constructed and fitted well with the pseudo-second-order equation. The results of adsorption isotherms show that the low temperature was conducive to adsorption. The results confirm that the purification of the natural compounds using macroporous resins followed by prep-HPLC was convenient and robust. Given that oxidative stress, inflammation and AGEs are important factors that endanger health, we have tried to establish a screening method for natural products with the three main effects of antioxidant, anti-AGEs and anti-inflammatory, which are conducive to screening out the potential bioingredients of complex Chinese herbs, thus laying the foundation for subsequent in-depth pharmacological studies. 

### 2.4. Bioassay In Vivo

#### 2.4.1. Pectolinarin Alleviates LPS-Induced Liver and Kidney Histopathological Injuries

Sometimes, good in vitro activity does not mean good in vivo activity. So, in the present research, in addition to the abovementioned in vitro tests, animal experiments were also conducted. As shown in Figure 3A,B, the content of linarin in the TFC fraction purified by the aforementioned prep-HPLC separation process is about 10%, and the remaining roughly 90% is pectolinarin. In addition, Kim et al. [30] has reported the protective effects of linarin against D-galactosamine and LPS-induced liver injury in mice. However, knowledge on pectolinarin’s hepatorenal protective effect is lacking. Herein, pectolinarin is subjected to successive in vivo bioassays. As shown in Figure 7A,B, pretreatment with high doses of pectolinarin (PH, 50 mg/kg) remarkably decreased the liver index of LPS-treated mice (*p* < 0.05); meanwhile, both high and low doses of pectolinarin can significantly decrease the kidney index (*p* < 0.01). Notably, the intervention effects of pectolinarin are as good as the positive drug dexamethasone (DEX, 10 mg/kg). Furthermore, we evaluated the effects of pectolinarin treatment on LPS-induced liver and kidney injuries by comparing the histological changes demonstrated by H&E staining (Figure 7C,D). The present study shows that the liver tissues of the control group had a normal hepatic architecture formed of cords of hepatocytes separated by hepatic sinusoids (400×). In contrast, the LPS group of liver sections showed abnormal histopathological appearances. The liver sections of mice treated with LPS demonstrated many histopathological features represented by the blurred liver lobules in the outline, the disordered arrangement of the hepatic cord, the swollen cytoplasm, vacuolation, necrosis, and the inflammatory infiltrate changes of hepatocytes. Dexamethasone (DEX, 10 mg/kg) prevented inflammatory liver damage, and the administration of pectolinarin (25 and 50 mg/kg) markedly inhibited vacuolation in LPS-induced liver injuries similarly to the DEX treatment. Histopathological results show that pectolinarin has obvious protective effects on LPS-induced liver injury. Similarly, some remarkable kidney pathological anomalies were found in the LPS-challenged mice, characterized by edema of renal tubular epithelial cells, tubular dilatation and distortion, brush border loss, infiltration of inflammatory cells, as well as tubular cell vacuolation. Notably, treatment with pectolinarin (50 mg/kg) or DEX (10 mg/kg) improved the anomalous histopathological alterations in mice challenged with LPS (Figure 7). Collectively, our results prove that pectolinarin could relieve the liver and kidney stress challenges posed by LPS. 

#### 2.4.2. Biochemical Assessment

In addition to the solid histopathological evidence, shown in Figure 8A, pectolinarin and DEX intervention significantly decreased the liver AST level in LPS-challenged mice (*p* < 0.01). Meanwhile, compared with the LPS group, pectolinarin groups showed significantly reduced levels of serum creatinine in mice (*p* < 0.05) (Figure 8B). Clinically, elevated serum creatinine levels reflect kidney dysfunction. Furthermore, the pathogenesis of LPS-induced injuries is related to the elevation of proinflammatory cytokines such as TNF-α [31]. Shown in Figure 8C, LPS led to a significant increase in the serum TNF-α level compared with the normal group (*p* < 0.05), but this was significantly reduced by the pectolinarin treatment (*p* < 0.001) in a dose-dependent manner, and these effects are as good as the positive drug dexamethasone (DEX, 10 mg/kg). All these results suggest that pectolinarin can improve LPS-induced liver and kidney damage.

### 2.5. Metabolomics Analysis

Combining the above results of in vitro and in vivo tests, pectolinarin should be a promising natural compound with antioxidant, anti-AGEs and anti-inflammation bioactivities. However, the underlying mechanisms of its in vivo effects is unknown. Nowadays, metabolomics analysis is one popular method that can present some meaningful clues for mechanistic investigation, and both GC-MS- and LC-MS-based metabolomics analyses are frequently used. In fact, each method has its own advantages and disadvantages [32]. Of course, if an integration of GC-MS and LC-MS metabolomics analysis can be carried out for the same batch of samples, it will be much better [33]. However, under a series of conditions, it is not adequate to preform both GC-MS and LC-MS profiling for some bio-samples. Considering our previous work [23,34], we have made use of non-targeted GC-MS-based metabolomics analysis to investigate the underlying mechanisms. Representative GC-MS total ion current (TIC) chromatograms of the normal control group (NC), the lipopolysaccharides-challenged (LPS, 5 mg/kg) group, and the lipopolysaccharides-challenged and high dose of pectolinarin (PH, 50 mg/kg)-treated group are shown in Figure 9A. The principle component analysis (PCA) score plot (R^2^X = 0.709, Q^2^ = 0.378) shows a clear separation for the three groups (Figure 9B). Subsequently, the OPLS-DA score plots were used to identify different metabolites via the variable importance of the projection (VIP) values between the LPS group and PH group. As shown in Figure 9B, the OPLS-DA model was established with good fitness and prediction (R^2^X = 0.82, R^2^Y = 0.999 and Q^2^ = 0.946). Moreover, a permutation test with 200 iterations confirmed that the constructed OPLS-DA model was valid and not over-fitted, as the original R^2^ and Q^2^ values to the right were significantly higher than the corresponding permutated values to the left (R^2^ = 0.995, Q^2^ = −0.239). Differential metabolites (DMs) were picked if they met criteria related to multiple data processing methods (VIP >1 and *p* < 0.05). As show in Table 4, 35 DMs were identified, including 6 increased and 29 decreased in the PH group compared to the LPS group. Using the HMDB for classification, of the 35 DMs, about 39% were sub-grouped as amino acids, peptides, and analogs; about 33% were lipids and lipid-like molecules; about 11% were carbohydrates and carbohydrate conjugates; about 11% were benzene and substituted derivatives; about 3% were organic acids and derivatives; and about 3% were amines (Figure 9C). These DMs were primarily located in the cytoplasm (49%), extracellular region (31%), membrane (11%), mitochondria (6%), and lysosome (8%) (Figure 9D).

Based on the metabolomics analysis-generated DMs, pathway enrichment was performed to identify the pathways. Functional analysis showed that the differential metabolites identified are primarily involved in amino acid metabolism and lipid metabolism (Figure 10). In this analysis, we identified several pathways that may be operated in PH-treated acute liver and kidney injury mice. The five most significantly different pathways are: (i) alpha-linolenic acid metabolism, (ii) arachidonic acid metabolism, (iii) cysteine and methionine metabolism, (iv) glycine, serine and threonine metabolism, and (v) glycerolipid metabolism.

In the present study, lipid metabolism has been shown to be markedly ameliorated in pectolinarin-treated mice. Compared with the LPS group, there were significant alterations in the levels of oleic acid, linolenic acid, palmitoleic acid, and arachidonic acid in PH group mice serum. It is well known that arachidonic acid is a polyunsaturated essential fatty acid, and it can trigger inflammation by releasing inflammatory mediator metabolites such as prostaglandins, thromboxanes, and leukotrienes when it is stimulated by LPS [35]. Linoleic acid is a precursor of arachidonic acid, which can be converted into various lipid mediators involved in the regulation of immunity [36]. After pectolinarin intervention, linoleic acid metabolism and arachidonic acid metabolism were significantly altered, indicating that pectolinarin effectively alleviated lipid metabolism and directly or indirectly inhibited the production of related inflammatory factors and mediators, manifesting acute liver and kidney effects.

As the metabolomics results show that the serum levels of several amino acids were altered in the PH group, so the amelioration of pectolinarin on LPS-induced acute liver and kidney injury was clearly associated with the alterations in amino acid metabolism. In the present study, the cysteine and methionine metabolism and the glycine, serine and threonine metabolism are two well-enriched amino acid metabolic pathways. In the study, glycine was increased after pectolinarin intervention, which is a potent antioxidant used to scavenge free radicals. Glycine can limit the production of reactive oxygen species by inhibiting the activation of macrophages and play a key role in the antioxidative defense of liver cell [37]. The cysteine and methionine metabolic pathways are the primary replenishment pathways in mammals that provide raw materials for glutathione synthesis [38]. Compared with the LPS group, the methionine content was significantly reduced after pectolinarin administration, indicating that the pectolinarin activated the cysteine and methionine metabolic pathways, increased the consumption of methionine, and made it more available for the glutathione production pathway, thereby enhancing the body’s antioxidant capacity and exerting liver and kidney protective effects.

## 3. Materials and Methods

### 3.1. Materials and Reagents

The dried over-ground portion of *C. japonicum*, collected in Hubei province (batch No. 201027), was purchased from Anhui Guanghe Medical Technology Co., Ltd. (Bozhou, China) and authenticated by Prof. Zhong Li from Guangdong Pharmaceutical University.

Acetonitrile, methanol and ethanol were provided by Merck Chemicals (Darmstadt, Germany). Fluorescein sodium salt (FL), methylglyoxal (MGO), fructose, bovine serum albumin (BSA), 2,2-azobis (2-methyl-propionamidine) dihydrochloride (AAPH) and 6-hydroxy-2,5,7,8-tetramethylchroman-2-car-boxylic acid (Trolox^®^) were supplied by Macklin (Shanghai, China). Macroporous resins (D101, HPD-100, HPD-500, HPD-600, HP-20, DM130, NKA-9, and AB-8) were obtained from Beijing Inluck science and technology Co. Ltd. (Beijing, China). LPS from *Escherichia coli* O55:B5 was supplied by Sigma (L2880; Sigma, Saint Louis, MO, USA). Dexamethasone was supplied by Macklin (Shanghai, China).

### 3.2. Macroporous Resin Enrichment Process

#### 3.2.1. Preparation of the Crude Extract of *C. japonicum*

The *C. japonicum* (5.0 kg) was extracted twice with water (the ratio of solid to water was 1:20, kg·L^−1^) under reflux for 2 h each time, then the extracted solution was concentrated under reduced pressure, and finally the crude extracted solution was filled to a volume of 5 L with water, and the final concentration was equivalent to 1 g of raw material per milliliter of solution.

#### 3.2.2. Determination of Total Flavonoid Content

The total flavonoid content was measured as formerly described by Zhang [39]. In brief, 2 mL of the diluted extract was added to a 10 mL volumetric flask, and 0.4 mL of NaNO_2_ reagent (5%, *w*/*v*) was added. The mixture was stirred well then kept at room temperature for 6 min; then 0.4 mL of Al(NO_3_)_2_ (10%, *w*/*v*) was added and 6 min later, 4.0 mL of NaOH regent (1 mol·mL^−1^) was added. Finally, the mixed solution was filled to the volume of 10 mL with 70% ethanol and then kept for 15 min at room temperature. The absorbance of the solution was measured against a blank at 510 nm using a spectrophotometer (SOPTOP, Shanghai, China). The content of total flavonoids in *C. japonicum* (TFC) was calculated based on rutin as the reference substance.

#### 3.2.3. Screening of Macroporous Resin

The optimum macroporous resin was evaluated by the adsorption capacity and desorption capacity [40]. The above-mentioned eight kinds of macroporous resin were subjected to the adsorption/desorption test. Briefly, 2.00 g of pretreated macroporous resin was mixed with 10 mL crude extract of *C. japonicum* in a conical flask at room temperature for 24 h. Following the adsorption experiment, the macroporous resins were slightly washed twice with distilled water. In the subsequent process of desorption, the adsorbed macroporous resin was mixed with 10 mL of 70% ethanol and kept at room temperature for 24 h. The ethanol elution parts were collected to determine the content of TFC and to confirm the dynamic adsorptive properties of various types of resin.

The adsorption capacity:$$Q_{r}=\frac{C_{0}-C_{1}}{C_{0}}\times 100\%$$

The desorption capacity:$$D_{r}=\frac{C_{d}\times V_{d}}{\left(C_{0}-C_{1}\right)\times V_{0}}\times 100\%$$
where *C*_0_ and *C*_1_ are the initial and equilibrium concentration of crude extract, and *V*_0_ is the extract volume (mL). *C_d_* represents the concentration (mg/mL) of total flavonoids in the desorption solution and *V_d_* is the volume of eluent (mL) in desorption experiments.

#### 3.2.4. Adsorption Kinetics

A total of 2.00 g of the best resin was placed in a conical flask, and 20 mL of crude extract of *C. japonicum* was added; after that, 80 mL of water was added for dilution and then adsorbed on a constant-temperature oscillator. Next, 1 mL of the solution was collected at 0, 5, 10, 30, 45, 60, 120, 150, 180 and 360 min to measure the total flavonoid content and calculate the adsorption ratio [22]. The adsorption kinetics curve was drawn using the data obtained. The adsorption kinetic data were fitted by means of the quasi-first-order kinetics model and quasi-second-order kinetics model, respectively. 

#### 3.2.5. Adsorption Isotherms

A total of 1.00 g of the best resin was placed in a conical flask, and then different concentrations of diluents of crude extract of *C. japonicum* were added to oscillate and adsorbed on a constant-temperature oscillator at 20, 30 or 40 °C at 100 rpm [24]. The concentration of total flavonoids in the adsorption solution was determined by the above-mentioned method (Section 3.2.2), and the equilibrium adsorption capacity of the resin was calculated according to the formula.

### 3.3. Purification and Identification of Pectolinarin and Linarin

#### 3.3.1. HPLC Analysis

HPLC determination of the crude extract was performed on a DIONEX 3000 UHPLC system equipped with a diode array detector and a Chromelon^TM^ Chromatography Data System (Thermo Fisher, Dreieich, Germany). Chromatographic separation was performed on a Kromasil C_18_ column (Akzonobel, Sweden, 250 mm × 4.6 mm, particles size 5 μm) at a flow rate of 1.0 mL·min^−1^, and monitored at 330 nm. In this case, acetonitrile (A) and water added with 0.05% formic acid (B) were used as the mobile phases. The gradient profile was as follows: 0~15 min, 90%→86% of B; 15~20 min, 86%→76% of B; 20~33 min, 76%→74% of B; 33~54 min, 74%→43% of B, 54~59 min, 43%→32% of B; 59~65 min, 32%→10% of B.

#### 3.3.2. Purification of two Compounds by Prep-HPLC

After macroporous resin treatment of the crude extract of *C. japonicum*, its desorption solution was filtered with the 0.45 µm nylon filter membrane and then it was injected into the Newstyle semi-preparative HPLC system equipped with an NP7000 pump and an NU3000 UV/VIS detector (Hanbon Corporation, Huaian, China) to get the fraction of a mixture of pectolinarin and linarin [41]. A Dynamic Axial Compression column (50 × 650 mm, particles size 10 μm) (Hanbon Corporation, Huaian, China) was used for the preparative isolation. Mobile phase A was 0.05% acetic acid aqueous solution and mobile phase B was methanol. The gradient elution program was as follows: 0~17 min, 5%~56% B; 17~21 min, 56%~62% B; 21~32 min, 62%~95% B. The flow rate was 70.0 mL·min^−1^. The injection volume was 8 mL, and the detection was monitored at 254 nm and 330 nm. After concentrating the objected fraction, a Luna C18 column (10 × 250 mm, particles size 10 μm) (Phenomenex Corporation, Torrance, CA, USA) was used to separate pectolinarin from linarin. The mobile phase was composed of methanol (A) and 0.05% acetic acid aqueous solution (B), using an isocratic elution of 50% A from 0 to 25 min. The flow rate was 4.0 mL·min^−1^, and the injection volume was 1 mL. The collected fractions were concentrated and then lyophilized to obtain purified pectolinarin and linarin powder.

#### 3.3.3. Identification of Pectolinarin and Linarin

The target compounds purified from the *C. japonicum* were confirmed by HR-MS, ^1^H- and ^13^C- NMR spectra. ESI-MS spectra were analyzed by an Agilent 6545 Series quadrupole time-of-flight (Q-TOF) mass analyzer equipped with an Agilent jet stream electrospray ionization (AJS- ESI) source (Agilent, Santa Clara, CA, USA) in the negative ionization mode. The ^1^H- and ^13^C-NMR spectra were obtained on a Bruker Avance 500 MHz (Bruker Corporation, Billerica, MA, USA).

### 3.4. Bioassay In Vitro

#### 3.4.1. Antioxidant Activity

Antioxidant activity was analyzed by oxygen radical absorbance capacity (ORAC) using a SpectraMax^®^-i3x Multi-Mode Microplate Reader (Molecular Devices, San Jose, CA, USA) with a 96-well plate [42]. Briefly, the pectolinarin and linarin were prepared in a 2 mg·mL^−1^ stock solution with absolute ethanol and diluted with sodium phosphate buffer (10 mM, pH 7.4) to different concentrations (10, 20 and 30 μg·mL^−1^). The solutions of AAPH (240 mM), fluorescein sodium (FL, 10 nM) and different concentrations of Trolox^®^ (200, 100, 50, 25 and 12.5 μM) were prepared in sodium phosphate buffer (10 mM, pH 7.4). The reaction mixture contained 150 μL of FL and 25 μL of sample per well. Trolox^®^ was regarded as the positive control and sodium phosphate buffer (10 mM, pH 7.4) was used as a blank. The plate was sealed and incubated in the dark for 30 min at 37 °C without shaking. Then, fluorescence measurements (Excitation, Ex. 485 nm; Emission, Em. 520 nm) were taken every 90 s to determine the background signal. After 3 cycles, 25 μL of AAPH was added quickly. The test was resumed and fluorescent measurements were taken for up to 120 min. The final results were calculated by comparing the net areas under the fluorescein decay curves between the blank and the samples. The triplicate measurement was performed and ORAC values were expressed as Trolox equivalents (TE) per gram (µmol TE/g). The assay was performed in triplicate.

#### 3.4.2. Anti-AGEs Activity

The anti-AGEs activity was determined as Boisard et al. described [28]. BSA (90 mg/mL) and MGO (4.5 mM) were dissolved in 50 mM sodium phosphate buffer (pH 7.4) with 0.01% of NaN_3_. Samples and aminoguanidine (AG, used as a positive control) were dissolved and diluted to the concentration of 2 mg/mL with PBS, respectively. BSA (90 mg/mL) was mixed with MGO (4.5 mM) together with AG or samples in tubes. The reaction mixture was incubated at 37 °C for 7 days in the darkness. The fluorescence of samples was measured using a microplate reader at the Ex. and Em. maxima of 340 and 420 nm, respectively. The inhibition rate of AGEs formation was calculated as follows: The % inhibition of AGEs formation = (1 − fluorescence intensity (sample)/fluorescence intensity (blank of sample)) × 100. The method for the glycosylation reaction of BSA and fructose is similar to the above method, but the BSA concentration is 20 mg/mL, and the MGO is replaced by 90 mg/mL of fructose solution. 

#### 3.4.3. Anti-Inflammation and Molecular Docking

Anti-inflammation activity was analyzed by the COX-2 inhibition method using SpectraMax^®^-i3x Multi-Mode Microplate Reader with COX-2 Inhibitor Screening Kit (Beyotime Biotechnology, Shanghai) following the manufacturer’s instructions [43]. Briefly, the compound was incubated with the enzymes for 5 min at 37 °C, then the fluorescence value of each well was measured (Ex. 560 nm, Em. 590 nm). Celecoxib, a known COX-2 inhibitor, was used as the positive control. Finally, the inhibition activity of each sample was calculated by the following equation: Inhibition activity (%) = (F_1_ − F_2_)/(F_1_ − F_3_) × 100%.
where F_1_, F_2_ and F_3_ were the fluorescence value of the 100% enzyme activity control group, the sample group and the blank control group, respectively. 

The molecular docking procedure was performed with Discovery Studio software 2021 (BIOVIA). The three-dimensional X-ray crystallographic structures of COX-2 (PDB ID: 6V3R) were downloaded from the Protein Data Bank (https://www.rcsb.org/, accessed on 3 March 2022) [44]. For ligand preparation, the structure of pectolinarin and linarin was constructed using ChemDraw Professional 14.0 software (PerkinElmer, Waltham, MA USA), saved in SDF file format and minimized using Discovery Studio software. The protein structures were cleaned and inspected for errors, hydrogens were added, and the water molecules were deleted prior to molecular docking [45].

### 3.5. Bioassay In Vivo

#### 3.5.1. Animal Experiments

Animal study protocols were approved by the Institutional Animal Care and Use Committee of the Guangdong Pharmaceutical University (No. SPF2017623). Male C57BL/6 mice were supplied by Guangdong Medical Laboratory Animal Center. The mice were acclimated for one week before the experiments and fed with a standard diet under 23 ± 2 °C, 12:12 dark to light conditions. All of the procedures were in strict accordance with the guide for the Care and Use of Laboratory Animals published by the US National Institutes of Health. All mice were divided into five groups (n = 6~8 each group): (i) normal group (NC), (ii) model group (LPS), (iii) positive drug Dexamethasone group (10 mg/kg, DEX), (iv) pectolinarin low-dose group (25 mg/kg, PL) and (v) pectolinarin high-dose group (50 mg/kg, PH) [46]. Shown in Figure 11, the DEX group mice were intraperitoneally injected with dexamethasone daily for 3 days, and the PL and PH group mice were with pectolinarin. At the same time, the NC and LPS group mice received the same volumes of saline. Two hours after the 2nd administration, all mice except for the NC group were injected with LPS (5 mg/kg). At 24 h after the 3rd administration, mice were anaesthetized and sacrificed, and samples of liver and kidney tissue were collected. Mouse serum was isolated from blood by centrifugation at 3500 rpm at 4 °C for 10 min and stored at −280 °C for biochemical analysis and metabolic profiling. Liver and kidney tissues were fixed in 4% paraformaldehyde and embedded in paraffin for routine histopathological examination.

#### 3.5.2. Histopathological Evaluation and Serum Biochemical Analysis

Routine Hematoxylin and Eosin (H&E) stainings of liver and kidney tissue were carried out as described previously [34]. Pathological changes in the tissues were evaluated by light microscope. Meanwhile, the serum TNF-α level was quantified using commercial ELISA kits (Cusabio, Wuhan, China). The levels of tissue AST and serum creatinine were measured using the commercial assay kits according to the manufacturer’s instructions (NanJing JianCheng Bioengineering Inc., Shanghai, China). 

### 3.6. Metabolomics Analysis

Serum metabolomics analysis was performed as described previously on a 7890B-5977B GC-MS system (Agilent Technologies, Santa Clara, CA, USA) [23]. Briefly, all the GC-MS raw data were used for batch molecular feature extraction using the MassHunter_B.08 (Agilent Co., Ltd., Santa Clara, CA, USA). Then, the generated data were exported and subjected to multivariate analysis for screening the differential metabolites. Here, VIP (Variable Importance in Projection) values of >1.0 and *p* < 0.05 were set as a statistical threshold for discriminating significantly differential metabolites. Subsequently, all the differential molecules were submitted to elucidate molecular networks and/or possible biological functions using MetaboAnalyst 4.0 (https://www.metaboanalyst.ca/, accessed on 3 March 2022). For details, see Appendix A: non-targeted metabolomics analysis.

### 3.7. Statistics

All the values are represented as mean ± SD. Data analysis were performed using Graphpad Prism 6.0 software (GraphPad, San Diego, CA, USA); Student’s *t*-test was employed and *p* < 0.05 was considered to be statistically significant. Principle component analysis (PCA) and pair-wise orthogonal projections to latent structures discriminate analysis (OPLS-DA) were performed on SIMCA-P 13.0 software (Umetrics, Umeå, Sweden).

## 4. Conclusions

In this study, an innovative process for the effective purification and screening of natural products with three main effects, namely, antioxidant, anti-AGEs and anti-inflammation activities, was established. With the suggested process, the purity of the resulting pectolinarin and linarin reached approximately 97.39% and 96.65%, respectively. The in vitro bioassay showed that pectolinarin and linarin have excellent antioxidant, anti-AGEs and COX-2 inhibition bioactivities. Furthermore, animal tests proved that pectolinarin can alleviate acute liver and kidney injury in an LPS-induced mice model. The metabolomics analysis results showed that pectolinarin attenuates LPS-challenged liver and kidney stress through regulating the arachidonic acid metabolism and glutathione synthesis pathways. Collectively, our results demonstrate that pectolinarin is one promising candidate for preventing oxidant stress, AGEs formation and inflammatory conditions. The established process also provides meaningful suggestions for the development of some other natural products.

## Figures and Tables

**Figure 1 molecules-27-08695-f001:**
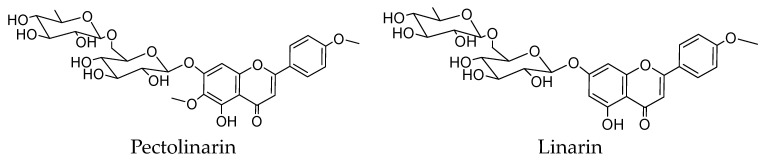
Chemical structure of pectolinarin and linarin.

**Figure 2 molecules-27-08695-f002:**
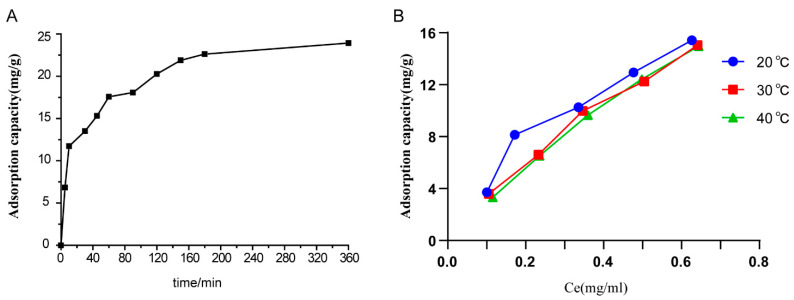
Adsorption behavior of TFC on microporous resin. (**A**) Adsorption kinetics curve of the TFC on D-101 resin. (**B**) Adsorption isotherms of the TFC on D101 resin at 20 °C, 30 °C and 40 °C. Ce, the absorption equilibrium concentration of analyte in the solution.

**Figure 3 molecules-27-08695-f003:**
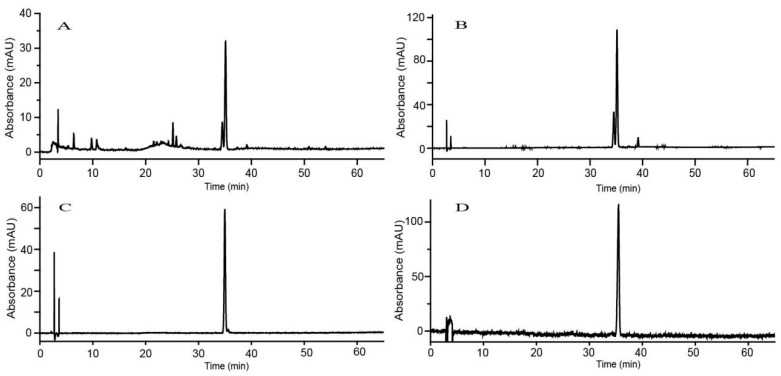
HPLC chromatograms of the TFC extracts before (**A**) and after (**B**) D-101 resin enrichment. HPLC chromatogram of purified linarin (**C**) and pectolinarin (**D**) detected at 330 nm.

**Figure 4 molecules-27-08695-f004:**
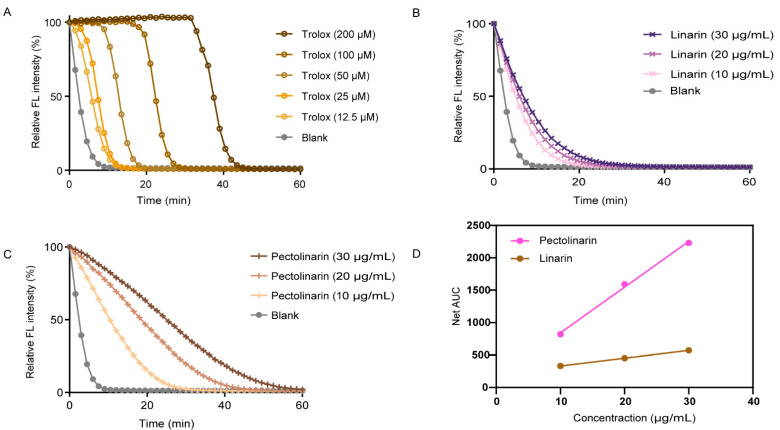
Antioxidant activity of pectolinarin and linarin illustrated by ORAC assay. (**A**) Representative kinetic curves from ORAC assay of varying concentrations of Trolox ranging from 12.5 µm to 200 µm. (**B**) Representative kinetic curves from ORAC assay of varying concentrations of pectolinarin ranging from 10 to 30 µg/mL. (**C**) Representative kinetic curves from ORAC assay of varying concentrations of linarin ranging from 10 to 30 µg/mL. (**D**) Linear relationship between Net AUC and concentration of pectolinarin and linarin.

**Figure 5 molecules-27-08695-f005:**
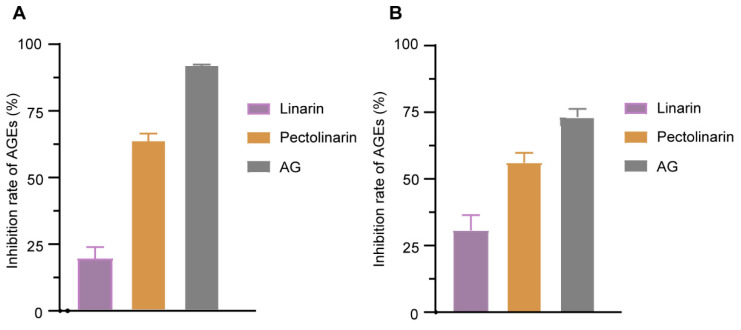
Pectolinarin and linarin suppress the AGEs formation. Anti-AGEs effects of pectolinarin and linarin on BSA-MGO-derived (**A**) and BSA-fructose-derived (**B**) advanced glycation end products formation at a concentration of 2 mg/mL. AGEs, advanced glycation end products.

**Figure 6 molecules-27-08695-f006:**
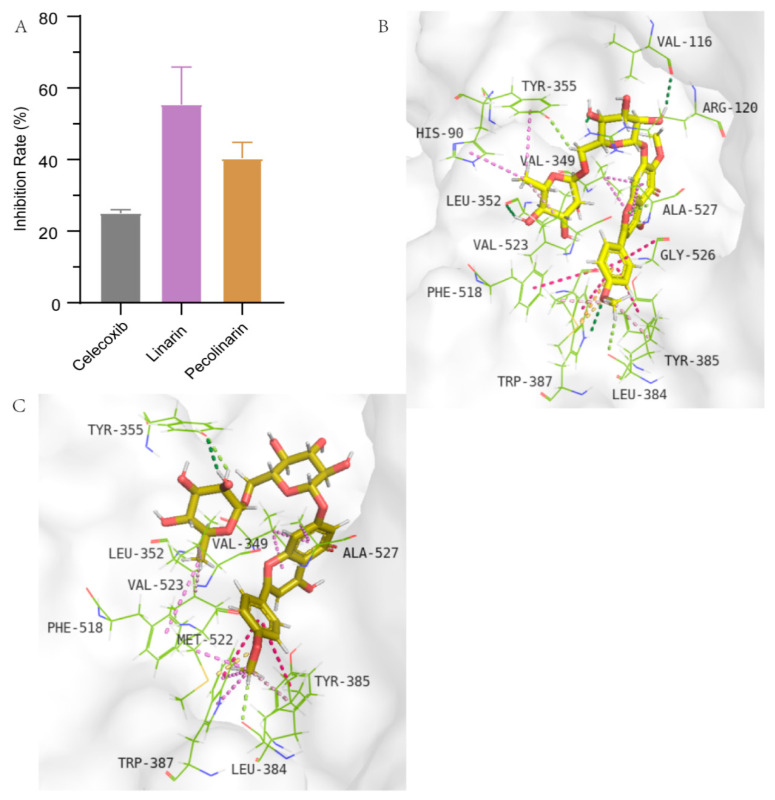
Results of inhibition assay and molecular docking analysis of linarin and pectolinarin with COX-2. (**A**) The inhibition rates of linarin (50 µg/mL), pectolinarin (50 µg/mL) and celecoxib (50 nM) on the activity of COX-2. (**B**) Pectolinarin binds in the cyclooxygenase active site of COX-2 (PDB code 6V3R). (**C**) Linarin binds in the cyclooxygenase active site of COX-2 (PDB code 6V3R).

**Figure 7 molecules-27-08695-f007:**
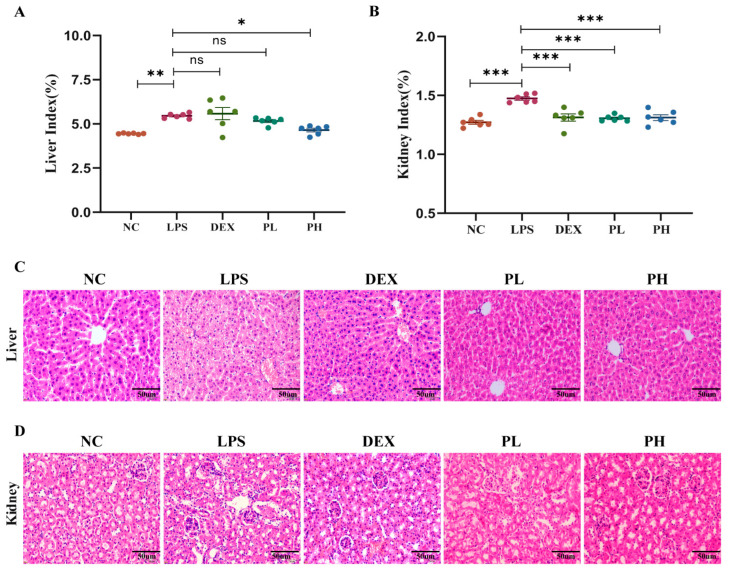
Pectolinarin alleviates LPS-induced liver and kidney histopathologic injury in mice. (**A**) Liver index. (**B**) Kidney index. (**C**,**D**) Representative histopathologic sections of liver and kidney tissue stained with H&E. Data are presented as mean ± SD (n = 6~8). * *p* < 0.05, ** *p* < 0.01, and *** *p* < 0.001, compared with LPS group. The kidney and liver indexes are calculated utilizing the following formula: kidney or liver weight/mouse body weight × 100%. NC, normal control group; LPS, lipopolysaccharides challenged (5 mg/kg) group; DEX, lipopolysaccharides-challenged and dexamethasone (10 mg/kg)-treated group; PL, lipopolysaccharides-challenged and low dose of pectolinarin (25 mg/kg)-treated group; PH, lipopolysaccharides-challenged and high dose of pectolinarin (50 mg/kg)-treated group. ns: not significant.

**Figure 8 molecules-27-08695-f008:**
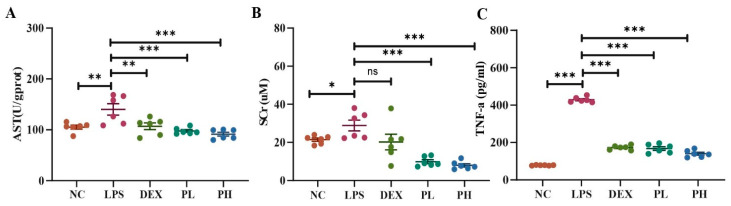
Pectolinarin attenuates LPS-induced liver and kidney metabolic stress. Pectolinarin treatment can significantly decrease the level of liver AST (**A**), serum creatinine (**B**) and serum TNF-α (**C**) compared with the LPS-challenged mice. Data are shown as mean ± SD (n = 6~8), * *p* < 0.05, ** *p* < 0.01, *** *p* < 0.001 compared with the LPS group. NC, normal control group; LPS, lipopolysaccharides-challenged (5 mg/kg) group; DEX, lipopolysaccharides-challenged and dexamethasone (10 mg/kg)-treated group; PL, lipopolysaccharides-challenged and low dose of pectolinarin (25 mg/kg)-treated group; PH, lipopolysaccharides-challenged and high dose of pectolinarin (50 mg/kg)-treated group. ns: not significant.

**Figure 9 molecules-27-08695-f009:**
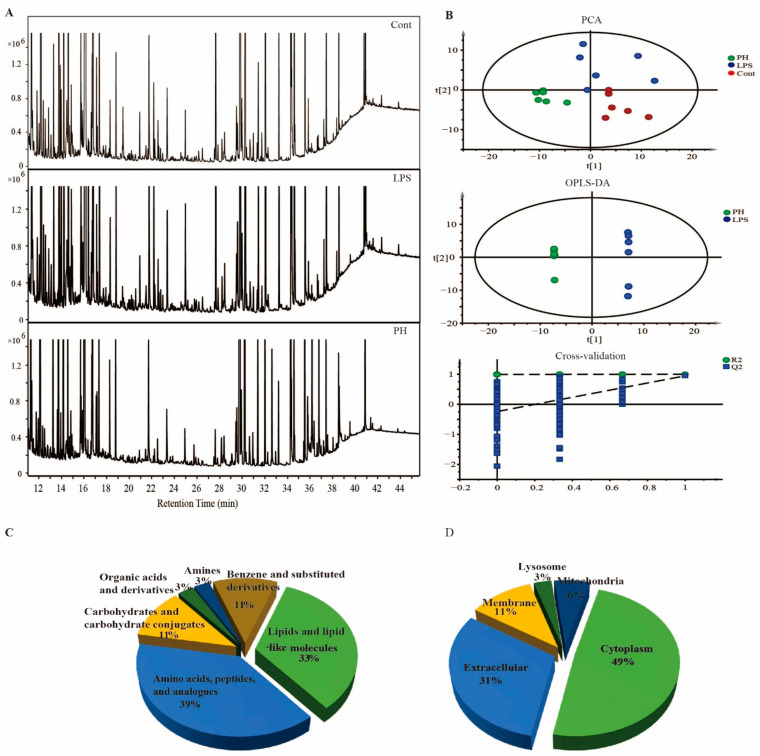
Non-targeted metabolomics analysis presented key differential metabolites between LPS-challenged and pectolinarin mice. (**A**) Representative GC-MS total ion chromatograph of serum from the control group (Cont), LPS group, and pectolinarin (50 mg/kg)-treated group (PH) mice. (**B**) Multivariate statistical analysis of GC-MS metabolic profiling data. PCA scores plot and OPLS-DA scores plot were derived from GC-MS spectra of three groups of mice (LPS, blue dot; Cont, red dot; PH, green dot) and statistical validation of the OPLS-DA model by permutation testing. (**C**) Chemical classification of the key differential metabolites based on the annotations of Human Metabolome Database (www.hmdb.ca, accessed on 3 March 2022) and their corresponding percentages. (**D**) Cellular locations of the key differential metabolites based on the annotations of the Human Metabolome Database (www.hmdb.ca, accessed on 3 March 2022) and their corresponding percentages. PCA, principle component analysis; OPLS-DA, pair-wise orthogonal projections to latent structures discriminate analysis.

**Figure 10 molecules-27-08695-f010:**
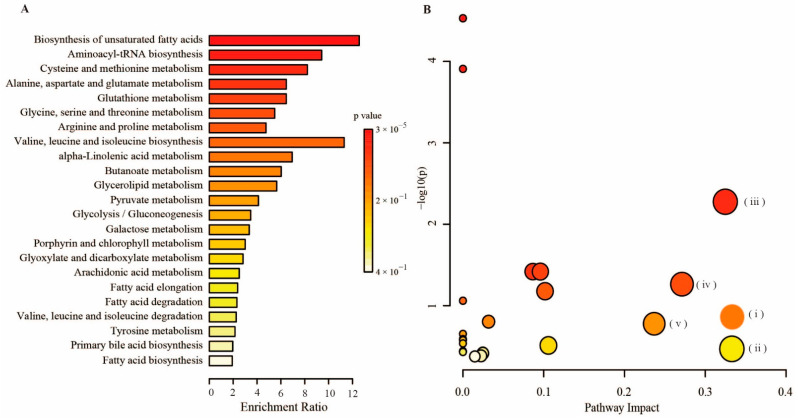
Summary of pathway analysis with MetaboAnalyst. (**A**) Enrichment analysis performed using the pathway-associated metabolites sets with MetaboAnalyst 4.0. (**B**) Overview of pathway analysis using Fisher’s Exact Test as algorithms with MetaboAnalyst 4.0; (i) alpha-linolenic acid metabolism, (ii) arachidonic acid metabolism, (iii) cysteine and methionine metabolism, (iv) glycine, serine and threonine metabolism, (v) glycerolipid metabolism.

**Figure 11 molecules-27-08695-f011:**
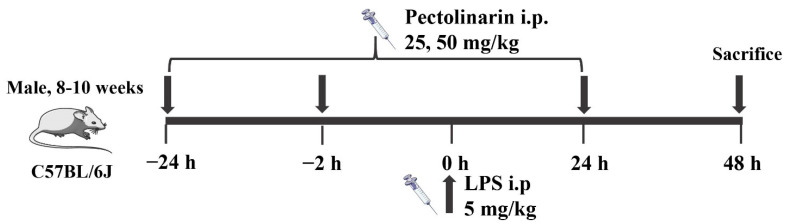
Timeline of the animal experiments.

**Table 1 molecules-27-08695-t001:** The static adsorption and desorption capacity of different resins on TFC.

Resin	Polarity	Adsorption Capacity (mg·g^−1^)	Desorption Capacity (mg·g^−1^)
HPD500	Polar	2.24 ± 0.08	1.68 ± 0.06
HPD600	Polar	1.83 ± 0.02	1.70 ± 0.06
NKA-9	Polar	1.92 ± 0.04	1.72 ± 0.40
DM130	Weak polar	1.78 ± 0.11	1.77 ± 0.13
AB-8	Weak polar	2.03 ± 0.07	1.82 ± 0.06
D-101	Non-polar	1.92 ± 0.03	2.04 ± 0.18
HPD100	Non-polar	1.05 ± 0.31	1.84 ± 0.11
HP-20	Non-polar	1.65 ± 0.14	1.68 ± 0.09

**Table 2 molecules-27-08695-t002:** Adsorption kinetics parameters for the TFC on the D-101 resin.

Kinetics Equations	Dynamic Parameters		
	*Q_e_ *(mg·g^−1^)	*K* _1_	*R* ^2^
Pseudo-first-order model	21.0497	0.0407	0.8768
Pseudo-second-order model	23.2981	0.0027	0.9493

**Table 3 molecules-27-08695-t003:** Langmuir and Freundlich isotherm parameters for adsorption kinetics of TFC on D-101 resin.

T/°C	Langmuir Model			Freundlich Model		
	*Q_m_* (mg·g^−1^)	*K_L_* (mL·mg^−1^)	*R* ^2^	*K_F_* ((mg·g^−1^)(mL·mg^−1^)^1/n^)	*n*	*R* ^2^
**20**	34.2411	0.0234	0.9938	21.2050	1.4208	0.9877
**30**	43.1711	0.0282	0.9921	21.2261	1.2949	0.9911
**40**	56.7479	0.0315	0.9992	21.9174	1.2003	0.9959

**Table 4 molecules-27-08695-t004:** Key differential metabolites detected by GC-MS for pectolinarin /LPS.

No.	Metabolite	RT (min)	*p*-Value (*t*-Test)	VIP Value	Fold Change *
1	Tyramine	12.04	3.50 × 10^−2^	1.38	−0.79
2	L-Lactic acid	12.21	4.60 × 10^−2^	1.16	−0.58
3	2-Phenylbutyric acid	12.40	2.40 × 10^−2^	1.47	−1.8
4	n-Butylamine	13.08	1.00 × 10^−2^	1.09	−2.6
5	4-Aminobutanoic acid	13.39	1.20 × 10^−2^	1.08	−0.87
6	L-alpha-Aminobutyric acid	13.98	3.30 × 10^−2^	1.27	−1.28
7	L-Isoleucine	14.87	4.20 × 10^−2^	1.31	−1.03
8	Acetylglycine	15.53	5.00 × 10^−3^	1.35	2.19
9	L-Asparagine	16.14	2.60 × 10^−2^	1.06	−0.54
10	Glycine	17.35	6.00 × 10^−3^	1.09	1
11	Cystathionine	17.55	1.50 × 10^−2^	1.01	−1.81
12	L-Norleucine	17.71	1.00 × 10^−3^	1.08	−1.64
13	L-Asparagine	18.40	5.00 × 10^−3^	1.38	−1.37
14	L-Methionine	19.24	3.30 × 10^−2^	1.55	−0.63
15	N-α-Acetyl-L-Lysine	20.32	4.00 × 10^−3^	1.09	−1.8
16	Glutamic acid	21.15	1.10 × 10^−2^	1.06	−0.43
17	Methyl alpha-D-galactopyranoside	21.45	1.20 × 10^−2^	1.25	−1.7
18	Benzoic acid	22.17	4.00 × 10^−3^	1.41	−3.66
19	D-Phenylalanine	22.23	1.80 × 10^−2^	1.09	−0.36
20	L-Proline	22.96	2.10 × 10^−2^	1.07	−0.87
21	4-Hydroxybenzoic acid	23.77	1.90 × 10^−2^	1.31	0.02
22	5,8,11-Eicosatrienoic acid	24.27	1.10 × 10^−2^	1.04	−1.19
23	Glycerol	30.50	3.00 × 10^−2^	1.35	−1.58
24	Palmitoleic acid	31.74	1.50 × 10^−2^	1.26	−1.96
25	Palmitic Acid	32.05	1.40 × 10^−2^	1.41	−1.22
26	Linolenic acid	34.3	2.00 × 10^−3^	1.33	−1.1
27	Oleic Acid	34.34	5.00 × 10^−3^	1.32	−1.08
28	Stearic acid	34.61	2.40 × 10^−2^	1.34	−0.78
29	Arachidonic acid	35.85	2.00 × 10^−3^	1.51	−0.79
30	9-Octadecenamide	35.95	0.00 × 10	1.18	2.68
31	Oleamide	36.21	6.00 × 10^−3^	1.22	3.26
32	Methyl galactoside	36.36	2.50 × 10^−2^	1.35	2.67
33	2-Palmitoylglycerol	37.22	3.70 × 10^−2^	1.54	−0.85
34	1-Monopalmitin	37.44	4.50 × 10^−2^	1.66	−1.68
35	Glycerol monostearate	38.55	2.50 × 10^−2^	1.69	−2.43

* Fold change was calculated as the logarithm of the average mass response (area) ratio between the two groups (i.e., fold change = log_2_[PH/LPS]).

## Data Availability

The data that support the findings of this study are available from the corresponding author upon reasonable request.

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
