# Peer review of "Purification Process and In Vitro and In Vivo Bioactivity Evaluation of Pectolinarin and Linarin from Cirsium japonicum"

_molecules, 2022, doi:10.3390/molecules27248695_

Round 1
Reviewer 1 Report
This manuscript was well-written and suitable for publication in molecules. The MS quality will benefit from minor corrections, appended below.
- Why was water used for the extraction of flavonoids in this study?
- TFC method showed be appropriately referenced
- In my opinion, when selecting a resin, a higher desorption rate of resin shouldn't just supersede the adsorption capacity. Moreover, Statistical analyses should be included. As I can see, NKA-9 and AB-8resin show similar Dr with D101. In fact, AB-8 could have been a better choice. More explanation should be provided!
- Authors should check the files uploaded (Supplementary and nonpublished files were duplicated)
- State the nmb replications used for experimental determinations
- Improve Keywords
-

Reviewer 2 Report
The authors evaluated "Purification process and in vitro and in vivo bioactivity evaluation of pectolinarin and linarin from Cirsium japonicum". Overall, the topic is interesting and time relevant. However, the main concern to accept this manuscript for publication is mentioned in the pdf file as attached.

Round 2
Reviewer 1 Report
The authors revised satisfactorily
Author Response
Thanks a lot for your comments and patience. Your constructive comments have helped us greatly to improve our manuscript.